# Reproducibility and Validity of a Food Frequency Questionnaire for Dietary Assessment in Adolescents in a Self-Reported Way

**DOI:** 10.3390/nu12072081

**Published:** 2020-07-14

**Authors:** Leyre Notario-Barandiaran, Carmen Freire, Manuela García-de-la-Hera, Laura Mª Compañ-Gabucio, Laura Torres-Collado, Sandra González-Palacios, Antonio Mundo, Marina Molina, Mariana F. Fernández, Jesús Vioque

**Affiliations:** 1Alicante Institute for Health and Biomedical Research, ISABIAL-UMH, 03010-Alicante, Spain; lnotario@umh.es (L.N.-B.); manoli@umh.es (M.G.-d.-l.-H.); lcompan@umh.es (L.M.C.-G.); l.torres@umh.es (L.T.-C.); sandra.gonzalezp@umh.es (S.G.-P.); 2Spanish Consortium for Research on Epidemiology and Public Health (CIBERESP), Instituto de Salud Carlos III, 28009 Madrid, Spain; cfreire@ugr.es (C.F.); marieta@ugr.es (M.F.F.); 3Instituto de Investigación Biosanitaria de Granada (ibs.GRANADA), 18012 Granada, Spain; antonio@alarconpsicologos.com (A.M.); molinalopezmarina7@gmail.com (M.M.); 4Department of Radiology, School of Medicine, and Center for Biomedical Research, University of Granada, 18071 Granada, Spain

**Keywords:** diet, nutrient intake, reproducibility, validity, food frequency questionnaire, dietary food record, adolescent, Mediterranean country

## Abstract

Tools to assess diet in a reliable and efficient way are needed, particularly in children and adolescents. In this study, we assess the reproducibility and validity of a semiquantitative food frequency questionnaire (FFQ) among adolescents in Spain. We analyzed data of 51 male adolescents aged 15–17 years from a prospective birth cohort study. Participants answered the FFQ twice in a self-administered way over a 12-month period. Reproducibility was assessed by comparing nutrient and food intakes from the FFQs, and validity by comparing nutrient intakes from the average of two FFQs and the average of two 24-Hour Dietary Recalls obtained in the period. Pearson correlation coefficients were calculated. The average of reproducibility correlation coefficients for food group intakes was 0.33, with the highest correlation for vegetable intake (r = 0.81); and the average for nutrient intake was 0.32, with the highest coefficients for α- and β-carotene (r = 0.65). Validity correlation coefficients ranged from 0.07 for carbohydrates to 0.53 for dietary fiber. The average of the validity correlation coefficients was r = 0.32. This study suggests that our FFQ may be a useful tool for assessing dietary intake of most nutrient and food groups among Spanish male adolescents in a self-administered way, despite reproducibility and, particularly validity, being low for some nutrients and food groups.

## 1. Introduction

Adolescence is a transitional stage of development from childhood to adulthood in which numerous biological and behavioral changes occur. Adolescents may be confronted with high-risk nutritional behaviors such as restrictive or excessive diets, or health problems such as overweight and obesity, most of them closely related to diet [1,2,3]. Thus, precise and accurate dietary assessment methods to investigate diet-disease relationships in adolescence are needed. In this regard, the food frequency questionnaire (FFQ) has been considered the most convenient dietary assessment method in most epidemiologic studies and populations, mainly due to the ease of administration, coding and processing (ref. 5 Willett). Consequently, FFQs have been validated in many different populations worldwide, mostly in adult populations [4,5], and though they have been used to explore diet-disease relationships in adolescents [2,6,7,8], the number of reproducible and valid FFQs for adolescents is still low [9,10]. Like in adults, FFQs in adolescents should also be validated for each specific population for which they will be used, due to factors that may vary depending on the geographical area, culture and age [11,12]. 

Although a few FFQs have been developed for adolescents [13,14,15,16,17], we are not aware of FFQs specifically validated for adolescents in Spain. In the context of the INfancia y Medio Ambiente (Environment and Childhood) Project, a multicenter population-based birth cohort study in Spain [18], we have previously developed and validated several FFQs for dietary assessment in pregnant women and their children at the ages of 4–5 and 7–9 years [19,20,21]. In accordance with the needs of the follow-up phases of the INMA study, the aim of this study was to evaluate the reproducibility and validity of a semiquantitative FFQ designed to assess diet in a self-administered way in male adolescents aged 15–17 years in the INMA Project, using two 24-Hour Dietary Recalls (24hDR) as the validation method. 

## 2. Materials and Methods

### 2.1. Study Population

We analyzed data from 51 male adolescents aged 15–17 years enrolled in the INMA-Granada cohort, one of the seven cohorts participating in the INMA Project. The INMA project is a population-based multicenter mother-child prospective cohort study designed to investigate the effect of environmental exposures and diet during pregnancy and childhood using a common protocol [18]. Briefly, 668 mother-son pairs were recruited at delivery between October 2000 and July 2002, and 152 children participated in the follow-up visit at the age of 15–17 years at the San Cecilio University Hospital (HUSC) of Granada [22]. The final sample for this validation study was based on 51 adolescents that agreed to complete all questionnaires in a self-administered way. This sample size provided the required statistical power to detect statistically significant correlation coefficients [12]. The scheme of the validation study is displayed in Figure 1. Participants received written guidelines to follow standardized procedures in order to complete the two FFQs and two 24hDRs in a self-administered way or with their parents’ help over a period of 9–12 months. All parents provided written informed consent. The study followed the principles of the declaration of Helsinki and were approved by the Ethics Committee of HUSC of Granada and Miguel Hernández University.

### 2.2. Dietary Assessment: Semi-Quantitative Food Frequency Questionnaire (FFQ)

We used a semiquantitative FFQ of 104 food items to assess the usual daily intake of food and nutrients. This FFQ was an adaptation of other FFQs previously validated, one among children aged 4 to 5 years [20], and another among pregnant women of the INMA Project [19]. These FFQs had a similar structure to the Harvard questionnaire [23]. To adapt the new FFQ to adolescents, we included food items and portion sizes appropriate for adolescents aged 15 to 17 years. This FFQ was designed to be self-administered by the adolescents although it was also recommended the parents help when needed, and in all cases, a final supervision was conducted by the interviewer during the visit at the hospital. Adolescents were requested to report the average frequency of consumption for the specified serving or portion size for each food item of the FFQ in the previous year. The questionnaire had nine possible frequency options, from “never or less than once a month” to “six or more times a day”. To estimate nutrient values and total energy intake, we used the published food composition tables of the US Department of Agriculture (USDA) and other published sources for specific Spanish food and portion sizes [24,25,26]. In order to estimate average daily nutrient intakes for each adolescent, we multiplied the frequency of consumption of each food item by the nutrient composition of the serving size specified in the FFQ and summed the results for all foods. Additionally, we estimated the mean daily intake of 17 food groups as shown in Table 1.

### 2.3. 24-Hour Dietary Recall (24hDR)

Two 24hDRs were collected from each adolescent in a self-reported way and in nonconsecutive days. The first 24h DR was completed when the first FFQ was completed. In order to stimulate the recall on the foods and beverages consumed in the previous day and to reduce the potential recall bias in fulfilling the 24hDR, adolescents were provided written instructions, based on the USDA Automated Multiple-Pass Method [27]. At total of 9–12 months after the first FFQ and 24hDR were completed, participants were also provided with the second FFQ and 24hDR, and were instructed to complete and return both questionnaires to the investigators by email or post mail (Figure 1). A reminder was sent to participants when a delay was noticed. 

A nutritionist performed the coding of all food items based on the recorded portion size, brand name and the method of food preparation as collected in the 24hDR. The Food Processor II ^®^ software was used to estimate the nutrient intakes. This software primarily uses the food composition tables from the USDA although we added information for specific Spanish foods available from Spanish food composition tables [24,25,26]. We used the average of two 24hDRs as the reference method to represent the individual intakes.

### 2.4. Statistical Analysis

All statistical analyses were performed using the statistical R software version 3.4.2 (R Foundation for Statistical Computing, Vienna, Austria; http://www.r-project.org). Means and standard deviations were calculated for nutrients and food groups from the FFQ and from the average of two 24hDRs. Student’s t-test was used to compare means of nutrients and food group intake between the two periods. Nutrient and food group intakes were log-transformed to reduce skewness and to improve their normality. Energy-adjusted intakes were computed using the residual method proposed by Willett [23], where each nutrient is regressed on total calories and the population mean is then added back to the calculated residuals. 

For the reproducibility analysis, we calculated Pearson correlation coefficients to compare nutrients and food groups from the two FFQs, the first completed at baseline and the second 9–12 months later. We also used Pearson correlation coefficients to evaluate validity of the FFQ, with the average of nutrient and food intakes of the two 24hDRs, that served as the reference FFQs (FFQav) with those of the average of the two 24hDRs (24hDRav) that we used as the reference method. We estimated correlation coefficients for log transformed intakes and energy-adjusted intakes that were also deattenuated to reduce the within-person variation found in the 24hDRs. The formula to calculate the deattenuated correlations was:$$\sqrt{1+\left\{\left(S2w/S2b\right)/n\right\}}$$
where S2w represents within-person variance and S2b between-person variance for each nutrient and *n* is the number of replicated 24hDRs, in our case *n* = 2 [23].

We also estimated Spearman correlation coefficients although the results were very similar to those observed for parametric correlations. Therefore, only Pearson correlations are presented. We calculated the percentage of agreement as the proportion of individuals who were classified correctly into the same or adjacent quintile for the reproducibility and validity analysis.

## 3. Results

The characteristics of participant adolescents in the validation study were very similar to nonparticipants although mothers from participant adolescents were two years older (Table 2). The mean age was 16.2 years and the mean body mass index 23.3 Kg/m^2^. The prevalence of overweight and obesity was 20.8 and 9.4% respectively. 

### 3.1. Reproducibility

The mean daily nutrient intakes and Pearson correlation coefficients between the nutrient intakes estimated by the two FFQ are presented in Table 3. Estimates of energy and nutrients intakes were in general slightly higher in the first FFQ than in the second FFQ, except for the intakes of β-cryptoxanthin, lycopene, vitamin B6 and vitamin C. Statistically significant correlations were observed for all log-transformed nutrients, ranging from r = 0.31 for sodium to r = 0.64 for trans-fatty acids. The average of log-transformed correlation coefficients was 0.43. When nutrient intakes were adjusted for total energy, correlation coefficients tended to decrease for most nutrients, ranging from r = 0.11 for zinc to r = 0.65 for α- and β-carotene; the average of energy-adjusted coefficient correlations was 0.36. Regarding the classification of nutrient intakes by quintiles as estimated by the two FFQs, the percentage of agreement (i.e., adolescents classified in the same or adjacent quintile) ranged from 58.5% for sodium to 79.2% for β-carotene; the average percentage of the agreements for nutrient intakes was 67.4%.

Regarding the reproducibility of the FFQ for food group intake, Table 4 shows the mean daily intake for the 17 foods and food groups estimated by the two FFQ. Most of the food intakes estimated by the first FFQ were higher than those estimated by the second, although only the intake of dairy products and sweet and sugar were statistically different (*p* < 0.05). White and red meat, fruit, potatoes and bread intakes estimated by the second FFQ were slightly higher than those estimated by the first. The correlation coefficients of food group intakes between the two FFQ showed a wider range of variation than that observed for nutrients, ranging from r = 0.10 for eggs and r = 0.81 for vegetables, although the average correlation coefficient was slightly lower than that observed for nutrients, r = 0.35 (energy-adjusted intakes).

### 3.2. Validity

Table 5 shows mean daily nutrient intakes and Pearson correlation coefficients from the average of two FFQs and two 24hDRs. Intakes from the FFQs were on average between 15 and 20% higher than that estimated by the two 24hDRs (*p* < 0.001). For the log-transformed intakes, the average of the correlation coefficient was 0.21, although this average increased to 0.28 when the analysis was based on energy-adjusted nutrient intakes. The magnitude of correlation coefficients was, in general, higher for energy-adjusted correlation coefficients ranging from 0.07 for total carbohydrates to 0.44 for magnesium. The average of deattenuated Pearson correlation coefficients was r = 0.32. The average percentage of agreement for validity was 59.4%, ranging from 51.0% (lycopene) to 68.6% (protein). 

Figure 2 shows the Bland–Altman plots for energy, total fat, trans fatty acids, and iodine. When we examined the Bland–Altman plots to explore the validity of the FFQ, the discrepancies between the two dietary assessment methods were equally likely in either direction, although for energy and total fat intake, the difference in absolute intakes increased with increasing average intake, with the mean of FFQ producing systematically higher estimates than the average of two 24hDRs.

## 4. Discussion

This study shows that the FFQ is a reliable method to assess diet in a self-administered way among male adolescents. The correlation coefficients of reproducibility for nutrient and food group intakes may be considered as moderate. The correlations for validity of the FFQ when compared with two 24hDRs were in general higher than 0.20 (average for deattenuated correlations, 0.32) which may be considered in a range between low and moderate validity when compared to other studies in adults [4,12]. However, to our knowledge, this is the first validation study of an FFQ to assess the nutrient and food intake among adolescents in Spain, and one of the few studies based on an FFQ for self-administered use. Although the sample size of the study was generally small, it was sufficient to detect statistically significant correlations of interest for validity (r > 0.20).

The Pearson’s correlation coefficients of reproducibility for most of the nutrients and foods groups were slightly higher than those obtained in a validation study assessing diet in a five-month period with children and adolescents in Australia where the average of correlation coefficients for reproducibility was 0.32 [28]. Other studies in adolescents have shown higher correlation coefficients in reproducibility than those shown in this study [29,30,31,32]. One reason for the higher correlations observed in those studies could be the much shorter interval used between the two FFQ administrations, which ranged from two weeks to one month. On the other hand, this study showed higher correlation coefficients compared to a study carried out in 52 adolescents in Peru in a six-month period [11], in which the average of correlation coefficients for reproducibility was 0.17. We used a 9–12-month period between the first and the second FFQ in order to control the potential influence of seasonal variability, although such a long period could also be a cause of poorer correlations due to changes in diet during that timeframe. Nonetheless, the range of reproducibility of correlation coefficients for most nutrients and food groups observed in this study may be considered according to Cade et al. [4] and Willett [12] as adequate, and it would suggest that our FFQ is an acceptably reproducible tool to assess the nutrient and food intake in Spanish male adolescents. 

The validity of this FFQ was assessed by comparing the nutrient intake estimates from the average of two 24hDRs with the average of the first and second FFQ. We used two 24hDRs as the most feasible reference method given the general characteristics of the study population, the available resources and because they have been also considered adequate to estimate the intake of energy and various nutrients [33]. Other alternative reference methods like the use of food diaries would have been more demanding and cumbersome for adolescent participants. FFQs tend to estimate higher nutrient intakes when compared to 24hDRs and Food Records [12] which has also been shown in validity studies with adolescents [28,29,30]. This was also shown in the current study, with the FFQ providing higher nutrient intake estimates than those from the 24hDRs for all nutrients. A possible explanation for the lower estimates from the 24hDRs could be that adolescents did not take into account all the foods eaten out-of-home, or the self-administered way of passing the 24hDR. On the contrary, it has been suggested that overestimation may also occur when FFQ with many food items are used [4], although this 104-item FFQ was similar to others we validated with adult Mediterranean populations in our area [19,34].

The correlation coefficient of validity in our study for energy intake between the FFQs and the 24hDRs was 0.28, higher than the observed in a study with 35 adolescents in the USA (r = 0.08) that used 3 day food records as a reference method [35]. The deattenuated correlations improved with respect to those based on energy-adjusted nutrient intakes for all nutrients due to correction of the within variation in nutrient intakes (average 0.32); the range for energy-adjusted correlation coefficients for validity observed for energy and nutrient intakes ranged from 0.07 to 0.44, higher than those reported by Rodriguez et al. [11], with their results ranging from −0.62 to 0.59, and within the range of 0.20 to 0.60 reported in a systematic review of dietary assessment methods for micronutrient intakes in adolescents [10]. The cross-classification of intakes by the two methods also showed a satisfactory level of agreement in nutrient and food intakes. Thus, the range of validity correlation coefficients observed for most nutrients in our study would suggest that the FFQ is also a reliable tool to assess diet and rank Spanish male adolescents according to their nutrient and food intake. 

The Bland–Altman diagrams showed no evidence of serious bias for any of the 25 nutrients examined, except for energy and total fat intakes, where some evidence that the FFQ overestimated intake compared with the 24hDR and that this overestimation increased with increasing average intake. The Bland–Altman diagrams only allow a graphical interpretation and the results should be considered together with other results (e.g., correlation coefficients, cross-classification into quantiles). Thus, our study shows an acceptable level of agreement in intake ranking between the two dietary methods used.

We acknowledge the limitations of our study. While 24hDRs have been considered an adequate reference method to validate the FFQ, they are not a perfect measure of the true dietary intake due to the high degree of intrapersonal variability. Moreover, the misreporting of energy intake is a common problem among adolescents [36]. In order to minimize these problems, parents and adolescents were given instructions to complete the 24hDRs following the USDA Automated Multiple-Pass Method to minimize under-reporting of consumed food and beverages due to memory lapse. Another limitation could be the use of a 9 to 12 month period, which could result in lower reproducibility and validity than desired, if real changes in diet occurred during that study period. However, we use a 9–12-month period to validate the FFQ in part to control seasonality. It should be also taken into account that the FFQ was validated only in male adolescents and this may be a limitation if the questionnaire was to be used in female adolescents. In addition, due to limited resources, we were not able to obtain more than two 24hDRs, including one weekend day, in order to control weekend variation. 

On the other hand, a major strength of the present study may be that we followed standardized protocols during the data collection of the INMA study fieldwork [18]. Although 51 participants represent a small sample size, it has been previously suggested that a sample size of 50 participants is acceptable (4). It should be noted that recruiting adolescents, especially older ones, is particularly challenging [37].

Moreover, the participants of the study were a subsample of a population-based birth cohort and the results could be more generalizable to adolescents of the same age range and gender.

## 5. Conclusions

This study suggests that our FFQ may be a reliable tool to assess diet in a self-administered way and rank Spanish adolescent males according to the dietary intake of a wide range of nutrients such as protein, dietary fiber, PUFA, omega 6, trans-fatty acids, alpha- beta-carotene, and also food groups such as dairy products, processed meat, fish, vegetables, fruits, cereals and sweets and sugar. The low validity observed for some nutrients should be further investigated in order to improve the overall validity of the FFQ and extend its use in studies with adolescents in Spain. In addition, testing the FFQ in a mixed-gender population is warranted.

## Figures and Tables

**Figure 1 nutrients-12-02081-f001:**
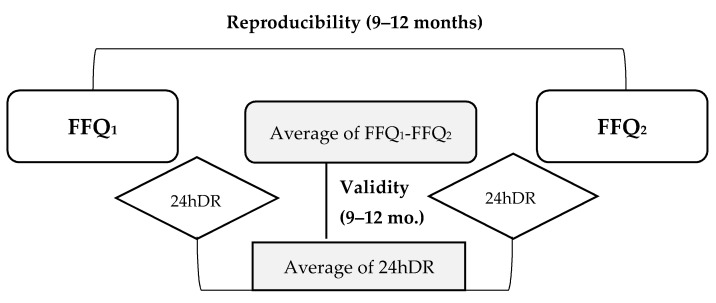
The design of the validation study among 51 adolescents aged 15–17 years of the INMA (Infancia y Medio Ambiente) project in Granada. FFQ, food frequency questionnaire; 24hDR, 24 hours dietary recall.

**Figure 2 nutrients-12-02081-f002:**
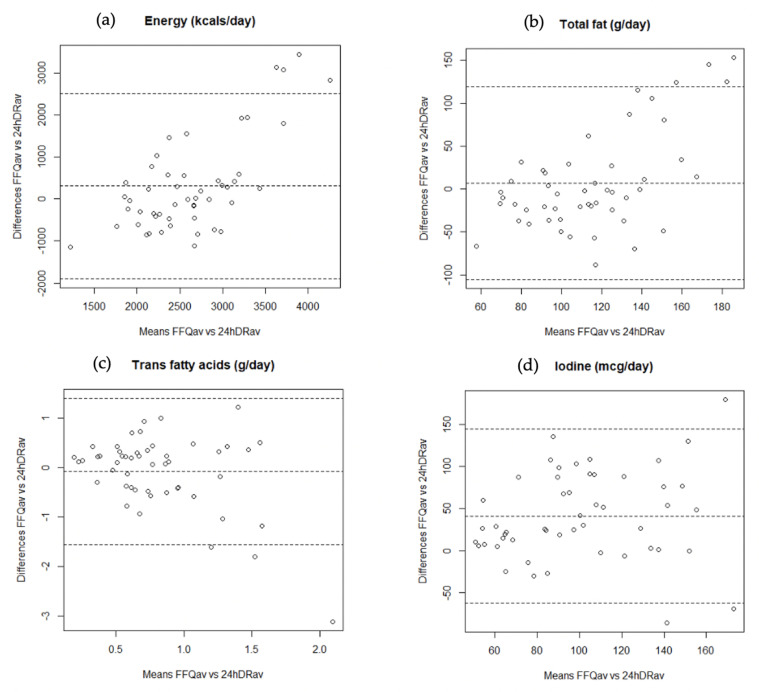
Bland-Altman plots showing the relationship between mean and differences in the daily intake of (**a**) energy; (**b**) total fat; (**c**) trans fatty acids, and (**d**) iodine estimated with the average of two FFQs and two 24hDRs.

**Table 1 nutrients-12-02081-t001:** Description of the food items integrated in the food groups.

Food Groups	Foods
Dairy Products	Whole dairy products; semi-skimmed, skimmed and fortified dairy products; Petit Suisse; cheese
Eggs	Eggs
White meat	Chicken or turkey
Red meat	Beef; pork or lamb
Processed meat	Sausages; ham; salami and others; serrano ham
White Fish	Swordfish; bonito and fresh tuna; small oily fish (mackerel, sardine, anchovy); canned sardine or mackerel
Blue Fish	An assorted or mixed fried fish (hake; gilthead sea bream and sole)
Seafood	Clams; mussels; squid; octopus; shellfish (crabs, shrimps, lobster); surimi and other fish-based food products
Vegetables	Raw vegetables (tomatoes; onions; lettuces; peppers and carrots) and cooked vegetables (spinach; cabbage; cauliflower or broccoli; carrots or squash; eggplant; zucchini or cucumber; green, red or yellow peppers)
Fruit	Oranges; other fruits (apples; bananas; pears; watermelon; melon; peach; kiwi; cherries; strawberries)
Nuts	Almonds; walnuts; peanuts and other types of nuts
Legumes	Lentils; chickpeas; beans; peas and green beans
Cereals	Breakfast cereals; corn; rice and pasta
Potatoes	Frozen French fry; homemade boiled/stew
Bread	White and whole breads
Sweets and sugar	Biscuits and baked goods; biscuits and baked goods with chocolate; peanut butter (e.g., Nutella/Nocilla); chocolate/cocoa powder; candies; jam and honey
Sweetened beverages sugary soft drinks	Packages juices; sugar soft drinks and artificially soft drinks

**Table 2 nutrients-12-02081-t002:** Characteristics of the adolescents of the Infancia y Medio Ambiente (INMA)–Granada cohort distinguishing between participants or nonparticipants in the validation study (Yes/No).

Characteristics	Participants (*n* = 51)	Non-participants (*n* = 101)	*p*-Value ^1^
	Mean (SD)	Mean (SD)	
Age (in years)	16.2 (0.4)	16.1 (0.4)	0.13
Body Mass Index (Kg/m^2^)	23.3 (5.0)	23.6 (4.9)	0.42
Mother’s age at delivery	41.1 (4.8)	39.1 (4.4)	0.02
	*n* (%)	*n* (%)	
Body Mass Index (Kg/m^2^), in categories			
Normal	37 (69.8)	68 (68.7)	0.85
Overweight	11 (20.8)	17 (17.1)	
Obesity	5 (9.4)	14 (14.1)	
Mother’s Educational Level			
Primary school	21 (40.4)	38 (38.8)	0.07
Secondary school	18 (34.6)	36 (36.7)	
University	13 (25.0)	24 (24.5)	

^1^*p*-values from Student’s t-test (continuous variables) and from Chi-square or Exact Fisher tests (categorical variables).

**Table 3 nutrients-12-02081-t003:** Mean daily nutrient intakes and Pearson correlation coefficients in two FFQs in adolescents aged 15–17 years of the INMA-Granada study (*n* = 51).

Nutrients Intake (Units/day)	FFQ1 ^1^	FFQ2 ^2^	*p*-Value ^2^	Pearson Coefficient (FFQ1 vs. FFQ2)		% of Agreement ^5^
	Mean (SD)	Mean (SD)		r ^3^	r adj. ^4^	
Energy (kcals/day)	3042 (1875)	2777 (1395)	0.412	0.43		62.3
Protein (g/day)	139 (90)	126 (74)	0.427	0.51	0.20	66.0
Total carbohydrates (g/day)	336 (183)	308 (145)	0.395	0.38	0.28	67.9
Dietary fiber (g/day)	29 (18)	26 (14)	0.396	0.35	0.26	62.3
Total fat (g/day)	131 (95)	119 (70)	0.462	0.48	0.24	66.0
SFA (g/day)	44 (36)	37 (23)	0.217	0.46	0.43	67.9
MUFA (g/day)	52 (36)	50 (30)	0.775	0.48	0.21	67.9
PUFA (g/day)	24 (18)	22 (14)	0.550	0.45	0.39	67.9
Omega 3 (g/day)	2.7 (2.3)	2.4 (1.8)	0.387	0.36	0.39	67.9
Omega 6 (g/day)	19.5 (15.6)	17.2 (12.5)	0.402	0.40	0.43	64.2
Trans fatty acid (g/day)	0.9 (0.9)	0.7 (0.6)	0.189	0.64	0.63	75.5
Retinol (µg/day)	1115 (1395)	644 (565)	0.026	0.39	0.28	75.5
α-Carotene (µg/day)	1112 (1392)	1053 (997)	0.804	0.58	0.65	73.6
β-Carotene (µg/day)	4681 (4115)	4220 (3160)	0.520	0.58	0.65	79.2
β-Cryptoxanthin (µg/day)	372 (273)	389 (265)	0.748	0.32	0.38	60.4
Lutein-Zeaxanthin (µg/day)	3434 (3649)	2674 (2131)	0.194	0.37	0.33	58.8
Lycopene (µg/day)	4761 (3543)	4946 (3190)	0.779	0.35	0.16	64.2
Folate (µg/day)	426 (269)	397 (198)	0.529	0.45	0.31	64.2
Vitamin C (mg/day)	179 (141)	200 (125)	0.415	0.50	0.55	73.6
Calcium (mg/day)	1600 (1023)	1357 (629)	0.144	0.44	0.45	75.5
Magnesium (mg/day)	447 (265)	398 (201)	0.284	0.39	0.27	66.0
Potassium (mg/day)	4506 (2542)	4037 (1921)	0.287	0.35	0.35	69.8
Sodium (mg/day)	4957 (3576)	4367 (2171)	0.308	0.31	0.27	58.5
Zinc (mg/day)	16 (10)	15 (8)	0.346	0.37	0.11	64.2
Iodine (µg/day)	135 (80)	111 (53)	0.077	0.30	0.40	66.0
**Average of correlation coefficients**				0.43	0.36	67.4

^1^ FFQ1 & FFQ2, the same FFQ was firstly administered at baseline (FFQ1) and secondly (FFQ2), between 9 to 12 months later; ^2^
*p*-value from paired t-tests; ^3^ r: coefficient correlations after nutrient crude intakes were log-transformed; ^4^ r adj: correlation coefficient using nutrient intakes adjusted for total energy; ^5^ Percentage of children classified in the same or an adjacent quintile in nutrient crude intakes; FFQ, food frequency questionnaire; SFA, saturated fatty acids; MUFA, monounsaturated fatty acids; PUFA, polyunsaturated fatty acids.

**Table 4 nutrients-12-02081-t004:** Mean daily food groups intakes and Pearson correlation coefficients in two FFQs in adolescents aged 15–17 years of the INMA-Granada study (*n* = 51).

Food Groups (g/day)	FFQ1 ^1^	FFQ2 ^1^	*p*-Value ^2^	Pearson Coefficients (FFQ1 vs. FFQ2)	
	Mean (SD)	Mean (SD)		r ^3^	r adj. ^4^
Dairy products	628 (424)	485 (235)	0.035	0.45	0.49
Eggs	27 (24)	27 (19)	0.921	0.19	0.10
White Meat	43 (49)	52 (107)	0.571	0.11	0.14
Red Meat	47 (41)	55 (67)	0.472	0.07	0.11
Processed Meat	120 (133)	93 (78)	0.203	0.40	0.52
White Fish	21 (19)	19 (20)	0.528	0.36	0.42
Blue Fish	42 (44)	40 (38)	0.820	0.33	0.35
Seafood	25 (40)	19 (33)	0.376	0.44	0.40
Vegetables	246 (244)	215 (167)	0.443	0.77	0.81
Fruit	376 (336)	432 (366)	0.418	0.38	0.44
Nuts	15 (23)	14 (31)	0.876	0.37	0.36
Pulse	77 (85)	54 (38)	0.078	0.11	0.13
Cereals	114 (83)	103 (108)	0.553	0.54	0.44
Potatoes	83 (83)	92 (105)	0.620	0.31	0.25
Bread	117 (67)	127 (94)	0.532	0.05	0.12
Sweets and sugar	69 (77)	42 (39)	0.023	0.39	0.44
Sweetened beverages	316 (304)	315 (422)	0.983	0.34	0.37
**Average of correlation coefficients**				0.33	0.35

^1^ FFQ1 and FFQ2, the same FFQ was firstly administered at baseline (FFQ1) and secondly (FFQ2), between 9 to 12 months later. ^2^
*p*-value from paired t-tests; ^3^ r, coefficient correlations after food groups intakes were log-transformed; ^4^ r adj, correlation coefficient using food groups intakes adjusted for total energy; FFQ, food frequency questionnaire.

**Table 5 nutrients-12-02081-t005:** Mean daily nutrient intakes and Pearson correlation coefficients from the average of two FFQs and two 24hDRs among children aged 15–17 years of the INMA-Granada study (*n* = 51).

Nutrients Intake (units/day)	FFQav ^1^	24hDRav ^2^	*p*-value ^3^	Pearson Coefficient			% of Agreement ^7^
	Mean (SD)	Mean (SD)		r ^4^	r adj. ^5^	r de-att. ^6^	
Energy (kcals/day)	2909 (1428)	2484 (466)	0.044	0.24			62.7
Protein (g/day)	133 (67)	106 (26)	0.009	0.18	0.40	0.40	68.6
Total carbohydrates (g/day)	322 (139)	270 (71)	0.020	0.25	0.07	0.07	58.8
Dietary fiber (g/day)	27 (12)	19 (6)	<0.001	0.36	0.38	0.53	62.7
Total fat (g/day)	125 (74)	111 (27)	0.217	0.15	0.17	0.24	58.8
SFA (g/day)	40 (26)	35 (10)	0.144	0.23	0.16	0.22	58.8
MUFA (g/day)	51 (29)	48 (13)	0.413	0.09	0.15	0.21	56.9
PUFA (g/day)	23 (14)	20 (8)	0.196	0.16	0.36	0.47	66.7
Omega 3 (g/day)	2.6 (1.7)	1.8 (1.0)	0.006	0.20	0.23	0.29	60.8
Omega 6 (g/day)	18.4 (12.1)	15.6 (6.9)	0.162	0.20	0.41	0.47	60.8
Trans fatty acid (g/day)	0.9 (0.7)	0.8 (0.7)	0.983	0.11	0.33	0.39	56.9
Retinol (µg/day)	879 (783)	621 (1416)	0.255	0.27	0.31	0.31	58.8
α-Carotene (µg/day)	1083 (989)	398 (763)	<0.001	0.30	0.40	0.46	62.7
β-Carotene (µg/day)	4451 (2929)	1213 (1423)	<0.001	0.24	0.37	0.52	52.9
β-Cryptoxanthin (µg/day)	381 (220)	173 (192)	<0.001	0.19	0.24	0.29	54.9
Lutein-Zeaxanthin (µg/day)	3054 (2179)	877 (566)	<0.001	0.23	0.39	0.39	54.9
Lycopene (µg/day)	4853 (2600)	3075 (2480)	0.001	0.13	0.24	0.29	51.0
Folate (µg/day)	411 (188)	229 (82)	0.451	0.35	0.30	0.30	58.8
Vitamin C (mg/day)	190 (111)	178 (566)	0.889	0.17	0.09	0.11	54.9
Calcium (mg/day)	1479 (722)	1116 (584)	0.006	0.32	0.22	0.22	58.8
Magnesium (mg/day)	422 (197)	315 (108)	0.001	0.25	0.44	0.44	64.7
Potassium (mg/day)	4271 (1821)	3176 (1143)	<0.001	0.21	0.34	0.34	62.7
Sodium (mg/day)	4662 (2435)	4534 (1484)	0.747	0.08	0.15	0.15	54.9
Zinc (mg/day)	15.6 (7.9)	12.2 (3.3)	0.006	0.15	0.15	0.15	60.8
Iodine (µg/day)	123 (55)	80 (39)	<0.001	0.22	0.33	0.33	60.8
**Average of correlation coefficients**				0.21	0.28	0.32	59.4

^1^ FFQav, average of FFQ1 and FFQ2; ^2^ 24hDRav, average of the two 24hDRs; ^3^
*p*-value from paired t-tests; ^4^ r, coefficient correlations after nutrient intakes were log-transformed; ^5^ r adj, correlation coefficient using energy-adjusted nutrient intakes; ^6^ deattenuated correlation coefficients after nutrient intakes were log-transformed and energy-adjusted; ^7^ percentage of children classified into the same or an adjacent quintile. FFQ, food frequency questionnaire; 24hDR, 24 hours dietary recall; SFA, saturated fatty acids; MUFA, monounsaturated fatty acids; PUFA, polyunsaturated fatty acids.

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
