# Peer review of "Reproducibility and Validity of a Food Frequency Questionnaire for Dietary Assessment in Adolescents in a Self-Reported Way"

_nutrients, 2020, doi:10.3390/nu12072081_

Round 1
Reviewer 1 Report
This is an interesting study assessing the reproducibility and validity of a Food Frequency Questionnaire for dietary assessment in male Spanish adolescents. The comments below are restricted to those which reflect suggested changes for improvement.
Abstract
Lines 33-34 among Spanish male adolescents
Line 34 validity being low
Keywords
Line 31 consider the terms “precision” (reproducibility) and “accuracy” (validity)
Introduction
Line 40 confronted with
Line 42 instead of efficient write “precise” or “reproducible”
Line 45 its low cost. Also provide a reference
Line 49 FFQs in adolescents should also be validated for each specific population they will be used.
Line 56 at the ages of
Line 58 in male adolescents
Line 59 Dietary Recalls
Line 59 as the validation method
Methods
Line 63 Long sentence. Break into two.
Line 70 provided the required statistical power to detect statistically significant correlation coefficients.
Line 81 and another among
Line 82 To adapt…. to adolescents, we
Line 83 appropriate for
Line 85 a final supervision was conducted by the interviewer
Line 86 to report the average frequency of consumption
Line 88 “never once” does not make sense, do you mean “never”?
Line 89 we used the published food composition tables of the USDA and other published sources…
Line 103 sent to participants
Line 104 A nutritionist
Line 107 Is there a Spanish database available? I assume that the list of foods in the US database would not be as relevant to the traditional Spanish foods
Line 113 calculated for nutrients and good groups
Line 114 Student’s t test was used to compare means of nutrient and food groups intake between the two periods
Line 118 population mean is then added back
Line 122 with the average of nutrient and food intakes of the two 24hDR, that served as the reference.
Line 131 Finally, we calculated (“we also” is being repeated twice in the last two sentences)
Results
Line 158 average correlation coefficient was
Discussion
Line 171 FFQ is a reliable method to assess diet in a self-administered way among male adolescents (not validated in females)
Line 178 Although the sample size of the study was generally small, it was sufficient to detect
Line 183 correlation
Line 191 during that timeframe
Line 192 of correlation
Line 194 in Spanish male adolescents
Line 195 nutrient intake estimates
Line 211 reference method
Line 219 the FFQ is also a reliable tool to assess diet in Spanish male adolescents
Line 226 to minimise under-reporting of consumed food and beverages due to memory lapse
Line 228 could result in lower reproducibility
Lines 231-234 Although, 51 participants represent a small sample size, it has been previously suggested that a sample size of 50 participants is acceptable [4]. It should be noted that recruiting adolescents, especially older ones, is particularly challenging.
Line 236 of the same age range and gender.
Line 238 FFQ is a reliable tool
Line 239 Spanish adolescent males
Line 241 In addition, testing the FFQ in a mixed-gender population is warranted.
Figure 1
The arrow from Average FFQs to Average 24hDR is confusing. 24hDR are not emerging from the FFQs, thus the arrow is not best describing the process. They are just compared, as were the FFQ1 and FFQ2, thus just use a line to denote comparison rather than showing direction with an arrow.
Also the timeframe (9-12 months) is shown for reproducibility but not for validity.
Table 1
White Fish small oily fish
Seafood correct “cramps” to crabs
Sweets and sugar
Sweetened beverages sugary soft drinks
Table 2
Title delete duplicate “of the” Infancia y Medio…
Mother’s age at delivery
Mother’s educational level
Define “primary” in the Spanish context
General comments
Limitations:
Mention in the limitations that the tool has not been validated in female adolescents.
Is there a more relevant database to use that the USDA one?
It is not mentioned if you accounted for weekend variation and seasonality.
Reviewer 2 Report
The authors report reproducibility and validity correlations from a study which estimated dietary intake from two FFQs and two 24 h dietary recalls in a small sample of Spanish adolescents. The study design is not new, however, results from a validation study are necessary in order to correct measurement errors inherent in FFQs, particularly if the authors intend to use the FFQ estimates in future studies. The manuscript is written well, overall; however there, are areas of concern in the abstract, methods, results, and discussion sections.
Decision: Revise
Minor concerns
ABSTRACT
Lines 32-33. "…our FFQ may be a useful tool for assessing dietary intake of most nutrients and food groups…" Not according to the Tables. The majority of the reliability and validity correlations are below the acceptable lower limit of 0.4.
METHODS
Line 71. r <= 0.20 represents poor validity. Validity correlations between 0.4 and 0.7 are considered acceptable, thus power analysis should have been performed using an effect size within the acceptable range of validity correlations.
Line 127. Should be S2w / S2b
Table 2. Should add another column showing the characteristics of the non-participants.
Tables 3-5. According to line 63, your sample included adolescents age 15-17 years, but your tables state age 14-15 years. Is this a typo? Please clarify.
Line 214. "0.07 to 0.44" Clarify if this range is for energy-adjusted or de-attenuated correlations.
Line 215. "0.20 to 0.60" Are these correlations inclusive of infants and children? Should compare correlations for adolescents only.
Line 234. According to your reference, a sample size of 50 is sufficient for Bland-Altman analysis; for correlations, 100-200 subjects are recommended.
Line 237. Conclusions do not reflect results presented in Table 5, which do not show correlations for foods. Perhaps state explicitly which nutrients have acceptable validity correlations.
Major concerns
METHODS
Lines 108-109. "We used the average of two 24hDR as the reference…" According to lines 86-87, the reference period for the FFQ is the "previous year". Ideally, the reference measure (in this case the repeated 24hDRs) should cover the same reference period of the FFQ. Therefore, the FFQ2 is the instrument that should have been validated. FFQ1 lacks appropriately timed reference measures. What is your justification for using the average of FFQ1 and FFQ2? Comment on the adequacy of two 24hDRs to represent intake of one year. Is this sufficient to capture the variation in nutrient and food intake in this sample?
